# SOX7: Autism associated gene identified by analysis of multi-Omics data

**Samantha Gonzales[1☯], Jane Zizhen Zhao[2☯], Na Young Choi[3], Prabha Acharya[3], Sehoon Jeong[4], Xuexia Wang[1]\*, Moo-Yeal Lee[3]\***

**1** Department of Biostatistics, Florida International University, Miami, Florida, United States of America, **2** Department of Psychology and Neuroscience, University of North Carolina at Chapel Hill, Chapel Hill, North Carolina, United States of America, **3** Department of Biomedical Engineering, University of North Texas, Denton, Texas, United States of America, **4** Department of Artificial Intelligence and Data Science, Sejong University, Seoul, South Korea

☯ These authors contributed equally to this work.
\* Moo-Yeal.Lee@unt.edu (M-YL); xuexwang@fiu.edu (XW)

## Abstract

Genome-wide association studies and next generation sequencing data analyses based on DNA information have identified thousands of mutations associated with autism spectrum disorder (ASD). However, more than 99% of identified mutations are non-coding. Thus, it is unclear which of these mutations might be functional and thus potentially causal variants. Transcriptomic profiling using total RNA-sequencing has been one of the most utilized approaches to link protein levels to genetic information at the molecular level. The transcriptome captures molecular genomic complexity that the DNA sequence solely does not. Some mutations alter a gene's DNA sequence but do not necessarily change expression and/or protein function. To date, few common variants reliably associated with the diagnosis status of ASD despite consistently high estimates of heritability. In addition, reliable biomarkers used to diagnose ASD or molecular mechanisms to define the severity of ASD do not exist. Therefore, it is necessary to integrate DNA and RNA testing together to identify true causal genes and propose useful biomarkers for ASD. We performed gene-based association studies with adaptive test using genome-wide association studies' (GWAS) summary statistics with two large GWAS datasets (ASD 2019 data: 18,382 ASD cases and 27,969 controls [discovery data]; ASD 2017 data: 6,197 ASD cases and 7,377 controls [replication data]) which were obtained from the Psychiatric Genomics Consortium (PGC). In addition, we investigated differential expression between ASD cases and controls for genes identified in gene-based GWAS with two RNA-seq datasets (GSE211154: 20 cases and 19 controls; GSE30573: 3 cases and 3 controls). We identified 5 genes significantly associated with ASD in ASD 2019 data ($KIZ\text{-}AS1$, $p = 8.67 \times 10^{-10}$; $KIZ$, $p = 1.16 \times 10^{-9}$; $XRN2$, $p = 7.73 \times 10^{-9}$; $SOX7$, $p = 2.22 \times 10^{-7}$; $LOC101929229$ also known as $PINX1\text{-}DT$, $p = 2.14 \times 10^{-6}$). Among these 5 genes, gene $SOX7$ ($p = 0.00087$) and $LOC101929229$ ($p = 0.009$) were replicated in ASD 2017 data. $KIZ\text{-}AS1$

**Data availability statement:** All relevant data are within the article and its supporting information files. Other publicly available data can be found at: https://pgc.unc.edu/for-researchers/download-results/, https://www.ncbi.nlm.nih.gov/geo/query/acc.cgi?acc=GSE211154, https://www.ncbi.nlm.nih.gov/geo/query/acc.cgi.

**Funding:** This study was financially supported by the National Institutes of Health (NCATS R44TR003491 and NIDDK UH3DK119982) and the University of North Texas (Startup). This research was supported in part by the National Institutes of Health (NIH) through the National Human Genome Research Institute (NHGRI) award for theFIU-Center for Genome Research (Award Number: UG3HG013615-01, contact PI: Xuexia Wang, MPI: Stephen Black). The content is solely the responsibility of the authors and does not necessarily reflect the official views of the NIH.

**Competing interests:** Authors have declared that no competing interests exist.

($p = 0.059$) and *KIZ* ($p = 0.06$) were close to the boundary of replication in ASD 2017 data. Genes *SOX7* ($p = 0.036$ in all samples; $p = 0.044$ in white samples) indicated significant expression differences between cases and controls in the GSE211154 RNA-seq data. Furthermore, gene *SOX7* was upregulated in cases than in controls in the GSE30573 RNA-seq data ($p = 0.0017$; Benjamini-Hochberg adjusted $p = 0.0085$). *SOX7* encodes a member of the SOX (SRY-related HMG-box) family of transcription factors pivotally contributing to determining of the cell fate and identity in many lineages. The encoded protein may act as a transcriptional regulator after forming a protein complex with other proteins leading to autism. Gene *SOX7* in the transcription factor family could be associated with ASD. This finding may provide new diagnostic and therapeutic strategies for ASD.

## Introduction

Autism spectrum disorder (ASD) is a heterogeneous grouping of neurodevelopmental traits which is diagnosed in roughly 1% of the world population [1]. ASD conditions are characterized by having attention-deficit hyperactivity disorder (ADHD), intellectual disability (ID), epilepsy, social communication deficits and restricted, repetitive, or unusual sensory-motor behaviors, or gastrointestinal problems [2]. A lot of research efforts have gone into understanding the causes of individual differences in autistic behavior. Twin and family studies strongly demonstrate that autism has a particularly large genetic basis, with estimated heritability ranging from 40% to 90% [3–6]. Molecular genetic studies revealed that the genetic risk for autism is shaped by a combination of rare and common genetic variants [7].

Over the past decade, genome-wide association studies (GWAS) and other type of genetic studies have identified increasing numbers of single nucleotide polymorphisms (SNPs) [8,9] and other forms of genetic variation that are associated with ASD [10]. It has been estimated that more than 100 genes and genomic regions are associated with autism [11,12]. While most of these studies focused on identifying heritable SNPs associated with ASD risk, other studies have demonstrated the influence of de novo mutations ranging from a single base [13,14] thousands to millions of bases long [15,16] to copy number variants (CNVs). Several likely gene-disruptive (LGD) variants in genes such as *GRIK2* [17] and *ASMT* [18] affecting autism-risk were found exclusively or more frequently in individuals with autism compared to control groups. Jamain et al. [19] showed strong evidence suggesting that mutations in *NLGN3* and *NLGN4* are involved in ASD. Additionally, deletions at Xp22.3 that include *NLGN4* have been reported in several autistic individuals. Roohi et al. [20] found out that *CNTN4* plays an essential role in the formation, maintenance, and plasticity of neuronal networks. Disruption of *CNTN4* is known to cause developmental delay and mental retardation. This report suggests that mutations affecting *CNTN4* function may be relevant to ASD pathogenesis. A review by Li and Brown [21] discussed a substantial body of evidence has resulted from genome-wide screening for several widely studied candidate ASD genes. Similarly, a large-scale international

collaboration was conducted to combine independent genotyping data to improve statistical power and aid in robust discovery of loci in GWAS [7]. This international collaboration also identified a significant genetic correlation between schizophrenia and autism with several neurodevelopmental related genes such as *EXT1*, *ASTN2*, *MACROD2*, and *HDAC4*. A combined analysis investigating both rare and common gene variants supported the evidence of the role of several genes/loci associated with autism (e.g., *NRXN1*, *ADNP*, 22q11 deletion) and revealed new variants in known autism-risk genes such as *ADPNP*, *NRXN1*, *NINL*, *MECP2* and identified new compelling candidate genes such as *KALRN*, *PLA2G4A*, and *RIMS4* [22]. Recently, Buxbaum [23] summarized the prevalence of some genetic variants in subjects ascertained for ASD.

Research investigating the gene expression profiles of those with ASD has also proven insightful genetic contributions to ASD. Expression levels of genes containing rare mutations associated with autism were evaluated in lymphoblasts from autism cases and controls, including aforementioned genes such as *NLGN3*, *NLGN4*, *NRXN1*, and *MECP2*. Out of these, *NLGN3* was found to be differentially expressed along with *SHANK3* [24]. More comprehensive gene expression analyses have confirmed susceptibility genes previously reported in GWAS-based analysis, identified novel differentially expressed genes, and biological pathways enriched for these genes [25]. RNA sequencing data analyses have elucidated several potential drivers of autism susceptibility, such as resting-state functional brain activity [26], dopaminergic influences in the dorsal striatum [27], overexpression of *FOXP1*, a gene involved in regulating tissue and cell type specific gene transcription in the brain [28,29], and genome-wide alterations to lncRNA levels, downregulation of alternative splicing events, and brain-region dependent alterations in gene expression [30]. Aforementioned studies indicate that integrating GWAS and RNA-seq data analysis can provide a better picture of the various underlying mechanisms behind a heterogeneous, multifaceted disorder like ASD.

In this study, we performed whole genome gene-based association tests for ASD with the adaptive test [31] using summary statistics from two large GWAS datasets which were obtained from the Psychiatric Genomics Consortium (PGC). We identified 5 genes significantly associated with ASD in ASD 2019 data. Among these 5 genes, gene *SOX7* was replicated in ASD 2017 data. Furthermore, two RNA sequencing data analyses indicated that gene *SOX7* was significantly upregulated in cases as compared to controls. *SOX7* encodes a member of the SOX (SRY-related HMG-box) family of transcription factors pivotally contributing to determining of the cell fate and identity in many lineages. The encoded protein may act as a transcriptional regulator after forming a protein complex with other proteins leading to autism.

## Materials and methods

### Datasets

**Discovery GWAS summary statistics:** The discovery dataset (labeled as asd2019) includes summary statistics from a meta-analysis of European samples derived from two cohorts: a population-based case control study from the Lundbeck Foundation Initiative for Integrative Psychiatric Research (iPSYCH) project and a family trio-based study from the Psychiatric Genomics Consortium (PGC) [8]. The iPSYCH samples included individuals born to a known mother and a resident of Denmark at the time of their first birthday. Cases were identified using the Danish Psychiatric Central Research Register, using diagnoses from 2013 or earlier by psychiatrist according to diagnostic code ICD10, which includes diagnoses of childhood autism, atypical autism, Asperger's syndrome, "other pervasive developmental disorders", and "pervasive developmental disorder, unspecified" [8]. The PGC samples consisted of 5 cohorts, whose trios were analyzed as cases and pseudo-controls. Details regarding these studies can be found in [8] and [7]. The combined sample size consisted of 18,382 cases and 27,969 controls. Imputation and quality control were performed *via* PGC's Ricopili pipeline, which ensures to produce robust, reproducible, and comparable datasets. The iPSYCH samples were processed separately in 23 genotyping batches, while the PGC samples were processed separately for each study. Genotype imputation was performed with IMPUTE2/SHAPEIT [32,33] in the Ricopili pipeline using the 1000 Genomes Project phase 3 dataset as the reference set. Regions demonstrating high linkage disequilibrium were excluded, and one of high similarity pairs of

subjects identified by PLINK's identity by state (IBS) analysis [34] were reduced at random, with a preference for retaining cases. Association was performed using PLINK on imputed dosage data and the meta-analysis was performed using METAL [8]. More detailed descriptions of each stage of the analysis can be found in Grove et al [8]. The summary statistics produced by this study and subsequently used for our analysis can be found at https://pgc.unc.edu/for-researchers/download-results/.

**Replication GWAS summary statistics:** The replication dataset (labeled as asd2017) includes summary statistics from a European-ancestry meta-analysis performed by the Autism Spectrum Disorders Working Group (AWG) of The Psychiatric Genomics Consortium (PGC), which aimed at improving statistical power to detect loci significantly associated with ASD. The meta-analysis was performed on data from 14 independent cohorts across different ancestries totaling over 16,000 individuals. For each step in the meta-analysis, each cohort was processed individually. Individuals were excluded if they were assessed at less than 36 months of age or if diagnostic criteria were not met from the Autism Diagnostic Interview-Revised (ADI-R) or the Autism Diagnostic Observation Schedule (ADOS) domain scores. While a "worldwide" meta-analysis on this aggregate dataset was performed, we derive our replication dataset based on the smaller European-only analysis consisted of 6,197 ASD cases and 7,377 controls [7]. Each stage of the imputation and quality control was performed similarly as the asd2019 data: Imputation and quality control on PGC samples were performed following the PGC's "Ricopili" pipeline. Since multiple studies were involved, necessary studies were performed to check for and remove duplicate individuals prior to imputation. Family trio-based data was organized as case and pseudo-controls. Criteria for SNP retention and other pre-imputation quality control steps can be found in the study's supplementary File 1 [7]. Genotype imputation was performed with IMPUTE2/SHAPEIT using the 2,184 phased haplotypes from the full 1000 Genomes Project dataset as the reference set. All 14 cohorts were tested for association individually using an additive logistic regression model in PLINK. More detailed information about each stage of the analyses performed by this study can be found in the study's supplementary File 1 [7]. The resulting summary statistics which were utilized in our analysis can be found at https://pgc.unc.edu/for-researchers/download-results/.

**RNA-Seq data of brain tissue (GSE211154):** Human postmortem brain tissue of each individual in a cohort of postmortem ASD cases and controls was obtained from the University of Maryland Brain and Tissue Bank, a brain and tissue repository of the NIH Neurobiobank. All ASD cases had confirmed diagnoses through Autism Diagnostic Interview-Revised (ADI-R) scores and/or received a clinical diagnosis of autism from a licensed psychiatrist. Controls were collected based on age and postmortem interval matched with each case [35].

RNA sequencing was conducted at the John P. Hussman Institute for Human Genomics Center for Genome Technology, University of Miami. Samples with RNA integrity (RIN) scores ≥4 were included for library preparation and sequencing. Total RNA was prepared using the Ovation SoLo RNA-Seq Library Preparation Kit (Tecan Life Science, Mannedorf, Switzerland). Sequencing was performed on the Illumina NovaSeq 6,000 (Illumina, San Diego, CA) with single end 100 bp reads targeting 25 million reads per sample. Overall, 39 samples (20 cases and 19 controls) were obtained for our analysis. S1 Table demonstrated the characteristics of the 39 samples. The study can be found in the Gene Expression Omnibus (GEO) database, under accession number GSE211154.

**RNA-Seq data of brain tissue (GSE30573):** The RNA dataset was obtained from a gene co-expression analysis which aimed to identify modules of co-expressed genes associated with ASD [36]. The study can be found in the Gene Expression Omnibus (GEO) database, under accession number GSE30573. Detailed descriptions of the raw data acquisition and quality control processes can be found in the supplementary information of [36] as well as the GEO accession viewer. Briefly, brain tissue samples (frontal cortex, temporal cortex, and cerebellum) were obtained from the Autism Tissue Project (ATP) and the Harvard Brain Bank. Cases were diagnosed using ADI-R diagnostic scores, which can be found along with other clinical data upon request from the ATP website. Total RNA was extracted from the sample tissues following the Qiagen miRNA kit instructions. Quality and concentration were assessed by Agent Bioanalyzer and Nanodrop, respectively. Reads were generated using Illumina GAII sequencer using manufacturer settings and were 73–76 nucleotides

in length. Raw sequencing data for the frontal and temporal cortex samples were available in the SRA run selector for 6 autism cases and 6 controls [36].

**eQTL data:** Top expression quantitative trait loci (eQTLs) for multiple tissues were obtained from the Gene-Tissue Expression (GTEx) portal, using the v8 release [37]. RNA-seq analysis was performed using STAR v2.5.3a for alignment and RSEM v1.3.0 for quantification. Genes were selected based on expression thresholds of >0.1 TPM in at least 20% of samples and ≥6 reads in at least 20% of samples. For each gene, expression values were normalized across samples using an inverse normal transformation. Genotype data was generated using whole genome sequencing for the subsequent eQTL analysis, which excluded variants with MAF < 1%. A total of 49 tissues across 838 donors were analyzed for eQTL associations as part of the GTEx experiments, however only 30 tissues contained a significant eQTL for SOX7. Cis-eQTL mapping was performed using FastQTL: full details regarding parameter specifications can be found on the GTEx portal website (https://gtexportal.org/home/methods). A majority of donors were white males aged 50–70. A full breakdown of donor characteristics can be found on the GTEx portal, under documentation (https://gtexportal.org/home/tissueSummaryPage#donorInfo).

## Quality control & preprocessing

**GWAS summary data:** After downloading the raw summary statistics from the PGC website, we performed quality control analysis to ensure robust and quality results. Only SNPs on autosomal chromosomes were used. First, SNPs with an imputation information metric (INFO) score below 0.9 were removed. Next, SNPs with strand-ambiguous alleles or non-biallelic loci were removed as well as SNPs with duplicate rs IDs. Z score was then calculated using each variant's odds ratio and standard error using the equation $Z = \log(OR)/SE(\log(OR))$. After quality control, the raw variants were sorted into hg19 RefSeq genes. Linkage disequilibrium (LD) within each gene was calculated using the 1000 Genomes European reference panel (phase 3): For each gene, a subset of the GWAS variants $\pm 1000$ *bp* of the gene's transcription start site and transcription end site were matched to the reference variants, ensuring both used the same reference allele and flipping Z score signs if necessary. Genes that contained less than 2 SNPs were removed. The Pearson's correlation between this subset of genotypes was calculated and used as the gene-wide LD. One SNP of a pair of SNPs with perfect correlation ($r_{ij} = 1$) within a gene was removed. The processed data was saved in 22 'RData' files (one for each chromosome) containing a list of data-frames, where each list element comprised of 1) SNP information for a specific gene and 2) its corresponding LD matrix.

**RNA-seq data of brain tissue (GSE211154):** Raw FASTQs were processed through a bioinformatics pipeline including adapter trimming by TrimGalore (https://githubcom/FelixKrueger/TrimGalore), alignment with the STAR package [35,38] to the GRCh38 human reference genome. Gene expression counts were quantified against the GENCODE v35 human gene release using the GeneCounts function in STAR. We downloaded gene expression counts directly from the GSE211154 webpage (https://www.ncbi.nlm.nih.gov/geo/query/acc.cgi?acc=GSE211154) [35].

**RNA-seq data of brain tissue (GSE30573):** The sequence read archive (SRA) accession list and associated sample metadata ("SRA Run Table") for GSE30573 were downloaded from the SRA run selector page for the study. Raw fastq files were downloaded from the SRA using the SRA Toolkit *via* the 'prefetch' and 'fastq-dump' commands [39]. We used FastQC to assess the quality of reads in each file, and MultiQC to visualize the results in batch format [40,41]. Only 1 sample failed the 'per sequence base quality' assessment and was subsequently trimmed of low-quality reads using the command-line tool 'fastq_quality_filter' from the FastX toolkit using a minimum quality Phred score of 20 and a minimum percent of bases per read to meet that threshold of 50% [42]. Reassessment *via* FastQC demonstrated this as sufficient trimming to meet the quality needed for downstream analysis.

After passing quality control, RNA-seq reads were aligned to a reference genome using STAR [38] by following two steps: genome indexing and the alignment to the indexed reference genome. We generated the genome index files using STAR's –genomeGenerate flag and setting –sjdbOverhang to 75 to match the maximum read length - 1 across the

samples. The reference genome FASTA file and corresponding annotation GTF file (GChr37/hg19, release 41) used to generate these index files were downloaded from GENCODE. After alignment, we used HTSeq [43] to estimate the number of reads per gene region (i.e., gene expression counts).

**GTEx eQTL data:** Preprocessing of downloaded eQTL data from GTEx (current release v8) was performed as part of the standard workflow in the R package "TwoSampleMR" [44,45]. Briefly, the top eQTL per tissue was matched to its corresponding SNP in the ASD2019 GWAS data, palindromic SNPs that could not be inferred via MAF as well as strand ambiguous SNPs were removed from the analysis. Additionally, effect direction was harmonized with the GWAS outcome data to ensure the same allele is referenced in both data sets.

## Statistical analysis

**Gene-based Association Test:** To perform gene-based association testing, we used the function 'sats' in the R package 'mkatr' [31]. This function computes p-values for 3 different gene based testing methods using GWAS summary statistics with an LD matrix calculated from a reference panel. A brief description of each method is as follows: Let $m$ denote the number of variants considered in a gene or gene region and let $(z_1, \cdots z_m)$ represent the GWAS summary statistics for that region. Let $R = (r_{ij})$ denote the estimated correlation matrix between Z statistics based on variant linkage disequilibrium (LD) calculated from a reference panel [31]. The tests included in the sats function are the sum test (a type of burden test), the squared sum test (a type of SKAT statistic) and the adaptive test (similar to the SKAT-O statistic). The three tests are as follows:

1. Sum test (ST): $B = \sum_{j=1}^{m} z_j$

2. Squared sum test (S2T): $Q = \sum_{j=1}^{m} z_j^2$

3. Adaptive test (AT): $T = \min_{\rho \in [0,1]} P(Q_\rho)$, *where* $Q_\rho = (1-\rho)Q + \rho B^2$ *and* $P(Q_\rho)$ denotes its p-value.

It can be shown that $Q_\rho$ asymptotically follows a weighted sum of independent chi-squared distribution with 1 degree freedom $[\chi^2(df = 1)]$ whose weights equal the eigenvalues of $R$, allowing for efficient computation of p value of $Q_\rho$, $P(Q_\rho)$. The minimum p-value of AT is searched for over a range of $\rho$ in the interval [0, 1] [31].

The ST is most valuable when all variants have the same direction of effect and approximately equal effect size for each genetic variants in the gene being tested, while the S2T will perform better than ST when genetic variants have different directions of effects. AT utilizes information from both ST and S2T, meaning AT can adapt to the genetic variants in the data better than ST or S2T alone. Indeed, the adaptive test shows the most robust performance across a wider range of scenarios [31]. Therefore, we report the results of AT in our gene-based genome-wide studies with GWAS summary statistics in this paper. More details regarding the derivation of these tests and their relation to the single-variant association test can be found in [31].

**Differential Gene Expression Analysis with GSE211154 data:** Gene expression of each gene was dichotomized as low expression (gene expression count ≤ median) and high expression (gene expression counts > median). We treated the binary variable of gene expression as a predictor variable and used multiple logistic regression model to conduct a gene expression differential analysis for each gene by adjusting for age at death, sex, race, and postmortem interval (PMI) on the basis of previous evidence [35] about the association of these variables with ASD (model 1).

$$\text{logit (p)} = \text{age at death} + \text{sex} + \text{race} + \text{PMI} + \text{GE} \tag{1}$$

In model 1, p is the probability of ASD; GE is the dichotomized gene expression count of each gene; PMI is the postmortem interval in hours. When the gene expression differential analysis is restricted to white participants, we excluded race from model 1.

**Differential Gene Expression Analysis with GSE30573 data:** For genes with gene expression counts at least 10, we used the R package DESeq2 [46] to perform differential gene expression analysis based on normalized gene expression counts. DESeq2 uses a generalized linear model to model the relationship between a trait and gene expression [46]. We used the Benjamini-Hochberg adjusted p-value to assess significance in gene differential analysis to control desired false discovery rate (FDR).

**Causal Inference with Two-Sample Mendelian Randomization:** To assess a potential causal effect of *SOX7* expression on ASD, we use the R package 'TwoSampleMR' to conduct 2-sample mendelian randomization using *SOX7* eQTLs obtained from GTEx as instruments [44,45]. Exposure and outcome data were harmonized prior to analyses to ensure the effect allele refers to the same allele in both datasets. We performed single-instrument analysis, using the Wald ratio method to estimate effects of *SOX7* expression in various tissues on ASD outcome. Thirty tissues were tested individually. A Bonferroni corrected threshold of $p \leq .0017$ was used to claim significance. Additionally, while the full GTEx data is the majority white donors (84.6%), the v8 release includes a separate, European only subset which contains 29 tissues with a significant *SOX7* eQTL. We performed a secondary analysis using this dataset, since our GWAS dataset is European only.

**Computing Environment:** RNA-seq data quality control, alignment, and counts were processed on the lonestar6 high-performance cluster provided by TACC at the University of Texas at Austin. Differential expression analysis and gene-based association tests were performed in a local Linux (Windows Linux Subsystem) environment using R (R-4.3.2) in RStudio. Two-sample mendelian randomization was performed in Windows 10, using R (R-4.3.3) in RStudio.

## Results

### Gene-based association test

**Discovery GWAS:** Out of approximately 19,000 genes tested for association with ASD, 5 genes were identified as significant with Bonferroni corrected p-values less than $p = 2.5 \times 10^{-6}$ (Fig 1 and Table 1). *SOX7* ($p = 2.22 \times 10^{-7}$) encodes a transcription factor involved in regulating embryonic development and cell fate determination [47]. *KIZ* ($p = 1.16 \times^{-9}$) encodes "Kizuna centrosomal protein", which plays a central role in stabilizing the pericentriolar region before the spindle formation step in cellular division [48]. A gene region which encodes long non-coding antisense RNA for *KIZ*, *KIZ-AS1* ($p = 8.67 \times 10^{-10}$) was also identified as significant, however the function of this antisense RNA has not been determined. *XRN2* ($p = 7.73 \times 10^{-9}$) encodes a 5'-3' exoribonuclease which is pertinent in promoting transcriptional termination [49]. Finally, *LOC101929229* ($p = 2.14 \times 10^{-6}$), also known as *PINX1-DT*, is a lncRNA that is considered a "divergent transcript" of the protein coding gene *PINX1*. While the divergent transcript function is not defined, *PINX1* encodes a protein that enables telomerase RNA binding and inhibitor activity and is involved in several related processes, including DNA biosynthesis and protein localization [50].

**Replication GWAS:** Among these 5 genes, gene *SOX7* (*p*=0.00087), and *LOC101929229* (*p*=0.009) were replicated in ASD 2017 data. Gene *KIZ-AS1* (*p*=0.059) and *KIZ* (*p*=0.06) were close to the boundary of replication in ASD 2017 data (Fig 1 and Table 1).

### Differential expression analysis in GSE30573

Among the five genes identified in the discovery of GWAS, gene *SOX7* (log2 Fold Change [LFC] = 1.17, $p = 0.0017$; Benjamini-Hochberg (BH) adjusted $p = 0.0085$), *LOC101929229* (LFC = 3.22, p = 5.83 ×$^{-7}$, adjusted p = 1.18 ×$^{-5}$), *KIZ* (LFC = 0.63, *p*=0.00099, BH adjusted *p*=0.0055) were also identified as significant in the differential gene expression analysis (Table 1). A comparison of case-control gene expression counts for *SOX7* can be found in Fig 2a, demonstrating that *SOX7* is consistently upregulated in autism cases compared to controls. The expression of *SOX7* is increased in autism patients relative to controls by a multiplicative factor of 2.25. In addition, the expression of *LOC101929229* is increased in autism patients than in controls by a multiplicative factor of 9.31.

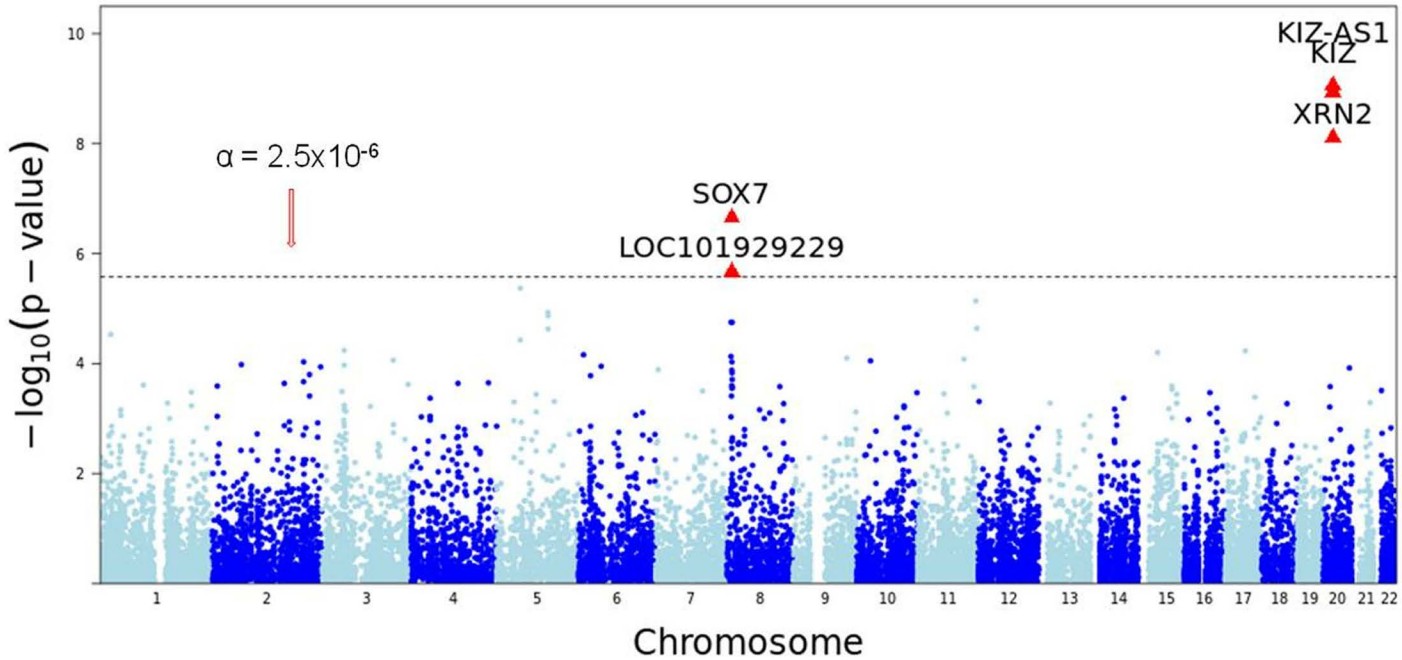

**Fig 1. Manhattan plot of a gene-based genome-wide association study for autism spectrum disorder in the discovery asd2019 GWAS data.**
Gene-based associated tests with the adaptive test are expressed as -log10 (p-value) on Y-axis. Chromosome 1–22 are labeled on X-axis. Each dot represents a gene tested for association with autism spectrum disorder; the dotted horizontal line represents a Bonferroni corrected p-value threshold of 2.5x10-6.

**Table 1. Significant genes identified in the gene-based test with GWAS summary data.**

| Gene | Chr. | Gene Based GWAS | | RNA-seq (GSE211154) | | | | | RNA-seq (GSE30573) (3 Cases/3 Controls) | | |
|---|---|---|---|---|---|---|---|---|---|---|---|
| | | *Discovery (asd2019)* | *Replication (asd2017)* | All Samples (20 Cases/19 Controls) | | White Samples (16 Cases/12 Controls) | | | | | |
| | | p-value* | p-value | Mean±std (Cases/Controls) | p-value | Mean±std (Cases/Controls) | p-value** | LFC | p-value | Adj. p-value |
| *SOX7* | 8 | 2.22E-07 | 0.0009 | 25±16.2/17.4±9.7 | **0.036** | 25.1±18.1/13.5±7.9 | **0.044** | 1.17 | **0.0017** | **0.0085** |
| *PINX1-DT* | 8 | 2.14E-06 | 0.009 | 5.9±7.6/9.8±9.2 | 0.19 | 6.5±8.4/8.8±9.0 | 0.60 | 3.22 | **5.83E-07** | **1.18E-05** |
| *XRN2* | 20 | 7.73E-09 | 0.10 | 475.2±169.7/428.5±124.1 | 0.61 | 499.7±180.1/419.9±129.9 | 0.94 | 0.34 | **0.001** | **0.007** |
| *KIZ* | 20 | 1.16E-09 | 0.06 | 1041.1±261.7/1028.4±278.8 | 0.17 | 1008.4±283.3/933.5±232.5 | 0.42 | 0.63 | **0.001** | **0.006** |
| *KIZ-AS1* | 20 | 8.67E-10 | 0.059 | 1.0±1.9/0.2±0.5 | 0.31 | 1.1±2.1/0.1±0.3 | 0.14 | 0.06 | 0.43 | 0.58 |

Abbreviation: Chr.: chromosome; LFC: Log2 fold change; adj. p-value: Benjamini-Hochberg adjusted p-value. *PINX1-DT* is also known as *LOC101929229*.

Notes: *p-value of gene based GWAS was obtained from the adaptive test (AT). **p-value of RNA-seq data analysis was obtained from Z test in the multivariable logistic regression. Significance level in the discovery is 2.5x10-6. Significance level in the replication is 0.05. Bold font indicates significant in each corresponding study.

## Differential expression analysis in GSE211154

Among the five genes identified in the discovery of GWAS, only gene *SOX7* demonstrated significantly elevated expression in ASD cases than in controls no matter in all samples ($p = 0.036$) or white samples (p = 0.044) (Tables 1 and 2). A

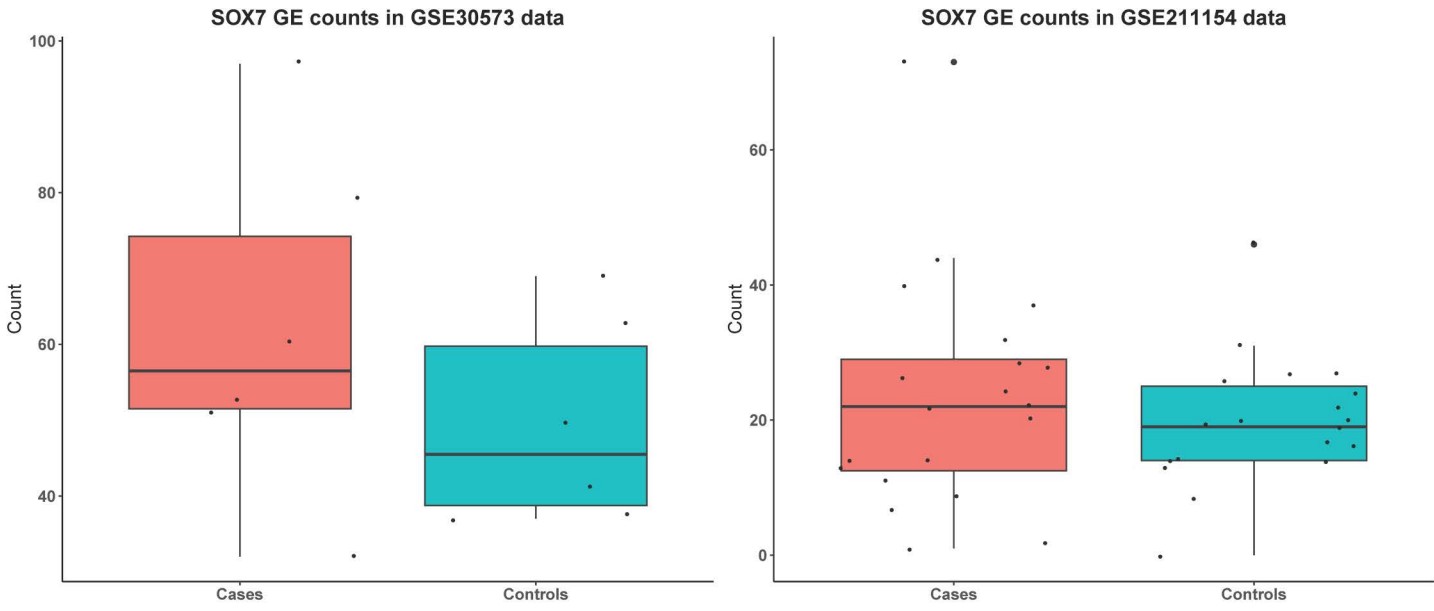

(p=2.22E−07 ASD2019; p=0.0017 ASD2017)

**Fig 2.** a. Comparison of *SOX7* gene expression between cases and controls in GSE30573 RNA-seq data. b. Comparison of *SOX7* gene expression between cases and controls in GSE211154 RNA-seq data.

**Table 2. Gene *SOX7* expression differential analysis with GSE211154 RNA-seq data.**

| Variable | β coefficient | Std. error | Z test | p-value* |
|---|---|---|---|---|
| **All samples** | | | | |
| Age at death (Year) | -0.01 | 0.04 | -0.23 | 0.82 |
| Postmortem interval (hour) | 0.11 | 0.06 | 1.86 | 0.06 |
| Sex (Male) | 0.02 | 1.02 | 0.02 | 0.98 |
| Race (White) | 2.71 | 1.29 | 2.10 | **0.04** |
| *SOX7* | | | | |
| Low expression (counts ≤median) | reference | | | **0.036** |
| High expression (count>median) | 2.58 | 1.23 | 2.10 | |
| **White Samples** | | | | |
| Age at death (Year) | 0.02 | 0.06 | 0.39 | 0.70 |
| Postmortem interval (hour) | 0.27 | 0.12 | 2.24 | **0.03** |
| Sex (Male) | -2.59 | 1.88 | -1.38 | 0.17 |
| SOX7 | | | | |
| Low expression (counts ≤median) | Reference | | | **0.044** |
| High expression (count>median) | 2.46 | 1.22 | 2.01 | |

Note: *p-value of Z test for each predictor is obtained from multiple logistic regression for all samples and white samples, respectively.

comparison of case-control gene expression counts for *SOX7* can be found in Fig 2b, demonstrating that *SOX7* is consistently upregulated in autism cases compared to controls.

## Two-sample Mendelian randomization

Out of 30 tissues tested, *SOX7* expression in 6 tissues were statistically significant after correcting for multiple testing in the full dataset, including 3 subregions in the brain: cerebellar hemisphere ($\beta = 0.1054$, $p = 5.31 \times 10^{-4}$), hypothalamus ($\beta = 0.1799$, $p = 4.71 \times 10^{-4}$), and spinal cord ($\beta = 0.1997$, $p = 8.84 \times 10^{-4}$) (Table 3). Interestingly, in the European only subset, the cerebellar hemisphere remained significant ($\beta = 0.0989$, $p = 5.31 \times 10^{-4}$) (Table 4). Full results for both analyses can be found in S2 and S3 Tables, respectively. These results demonstrate a causal relationship between *SOX7* expression and ASD.

## Discussion

Through gene-based analysis, we identified 5 gene regions (*KIZ*, *KIZ-AS1*, *XRN2*, *LOC101929229*, and *SOX7*) significantly associated with ASD. gene *SOX7* and *LOC101929229* were supported by results from the replication study in a different GWAS data and the differential gene expression analysis performed on publicly available RNA-seq data.

   *KIZ* is located on chromosome 20, and encodes Kizuna centrosomal protein, which aids in stabilizing the pericentriolar region of centrosomes before spindle formation. *KIZ* has been identified as significantly associated with autism in previous GWAS [8], TWAS [51], gene based analysis [52], and methylation-based studies [53], and the involvement of cell cycle regulation in autism susceptibility has also been implicated in previous research [54,55]. *KIZ* has also been found to be a potentially shared genetic loci between ASD and attention-deficit hyperactivity disorder (ADHD), providing support for its involvement in neurological disorders [56].

   *XRN2* is located next to *KIZ* and encodes a 5'-3' exonuclease that is involved in myriad RNA management processes, including transcriptional termination, miRNA expression regulation, nonsense-mediated mRNA decay, and rRNA maturation [57–60]. *XRN2* has been found to play a role in regulating miRNA expression in neurons specifically, and altered miRNA expression regulation has been investigated as a potential mechanism for autism susceptibility [61–65]. Likewise, disruption of proper RNA metabolism as a result of altered expression of RNA binding proteins has been implicated in neurological disease as a whole, and the *XRN* gene family is involved in nonsense-mediated decay of mRNA, a process that has been implicated in autism pathophysiology [66,67]. Previous GWAS have reported SNPs in the region containing *XRN2* to be significantly associated with ASD, affirmed by gene-based analysis using MAGMA [8]. Additionally, a

**Table 3. Two-Sample MR between *SOX7* expression by tissue and ASD.**

| Tissue | $\beta$ coefficient | Std. error | p-value |
|---|---|---|---|
| Adrenal Gland | 0.2808 | 0.0567 | 7.31e-07 |
| Brain Cerebellar Hemisphere | 0.1054 | 0.0304 | 5.31e-04 |
| Brain Hypothalamus | 0.1799 | 0.0514 | 4.71e-04 |
| Brain Spinal cord cervical c-1 | 0.1997 | 0.0601 | 8.84e-04 |
| Lung | -0.5016 | 0.1290 | 1.01e-04 |
| Muscle Skeletal | 0.3941 | 0.1186 | 8.89e-04 |

**Table 4. Two-Sample MR between *SOX7* expression by tissue and ASD in EUR subset.**

| Tissue | $\beta$ coefficient | Std. error | p-value |
|---|---|---|---|
| Brain Amygdala | -0.1670 | 0.0423 | 7.74e-05 |
| Brain Cerebellar Hemisphere | 0.0989 | 0.0285 | 5.31e-04 |

transcriptome-wide association study (TWAS) found *XRN2* to be significantly upregulated in autism, in accord with our findings [68]. Another gene-based analysis found *XRN2* to be associated with ASD and upon further investigation *via* gene-network analysis and enrichment analysis found that not only does *XRN2* interact with several genes in the cAMP signaling pathway and RNA transport network, but that the enriched KEGG/GO terms for *XRN2* (spliceosome, RNA transport, and nucleic acid binding) found to be associated with ASD are also essential processes pivotal to early development [52]. The extensive involvement of *XRN2* in such complex mechanisms of gene expression regulation, particularly in neuronal cell types, offers possible insights into the vast heterogeneity of ASD and its overlap with other neurodevelopmental disorders. In fact, more recent research efforts have focused on ascertaining genetic commonalities between ASD and related disorders such as ADHD, obsessive compulsive disorder (OCD), and Tourette Syndrome, of which *XRN2* seems to be a shared significant locus [69,70].

*SOX7* is of particular interest due to its hallmark involvement in the regulation of Wnt/$\beta$-catenin pathway (Fig 3), an important developmental signaling pathway. *SOX7* and its related SOX family genes encode transcription factors that are critical to the downregulation of the canonical Wnt/$\beta$-catenin signaling pathway, which controls embryonic development, adult homeostasis, and is involved in a multitude of cellular processes [71,72]. While the Wnt pathway is ubiquitous to nearly all tissue types, proteins involved in Wnt signaling in the brain specifically have been found to localize in the synapses and influence synaptic growth, and knockout murine models of ASD risk genes that are a part of the Wnt pathway have provided support for the disruption of this pathway in autism-like behaviors [73]. Indeed, the Wnt/$\beta$-catenin signaling pathway has been suggested as a possible avenue for autism pathogenesis in several studies [73–79].

*SOX7* also regulates angiogenesis, vasculogenesis, and endothelial cell development, and the SOX family of transcription factors are critical to cardiovascular development [80,81]. For example, *SOX7* was found to be upregulated in sustained hypoxic environments, mediating angiogenesis [82], and a knockout model of *SOX7* was found to result in profound vascular defects, and demonstrated that *SOX7* has an essential role in vasculogenesis and angiogenesis in early development [83]. Links between *SOX7's* role in developmental delay and congenital heart disease have been investigated. Specifically, deletions in the region where *SOX7* resides have been demonstrated to simultaneously cause congenital heart defects and intellectual disability [84,85].

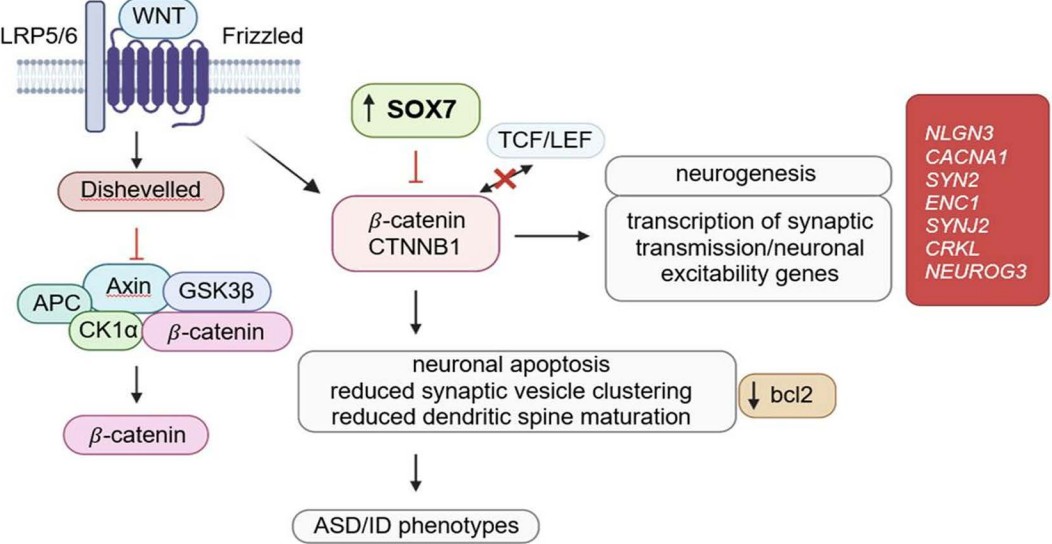

**Fig 3. Proposed pathogenesis of autism via *SOX7*.**

Additionally, Wnt signaling has been demonstrated to orchestrate differentiation of neural vasculature, such as the blood-brain barrier [86,87]. Likewise, there is evidence of vascular involvement in the development of autism [88–91]. One review in particular suggests that mutations affecting the delicate interactions between Wnt signaling and Shh pathways may alter blood brain barrier integrity in autism by aberrantly interacting with neurovascular molecules [92].

Lastly, oxidative stress has been researched as a potential source of autism susceptibility [93,94], and the interaction between altered vasculature and autism during oxidative stress could point to another potential source of pathogenesis [88]. Indeed, the role of Wnt/$\beta$-catenin signaling in oxidative stress has been implicated in autism susceptibility directly [95]. This combination of evidence that implicates both Wnt signaling and *SOX7* interactions in the multitude of interrelated processes that have been suggested as mechanisms behind the etiology of ASD, supplemented by our findings, provide ever-mounting support for more in-depth investigations of these particular genes and pathways.

Wnt/$\beta$-catenin, oxidative stress, and impaired/altered vasculature have all been implicated in the development of ASD. These three factors are involved with each other and multiple systemic processes, which may contribute to ASD's symptom heterogeneity. The fact that *SOX7* is involved in the regulation of Wnt/$\beta$-catenin and vasculogenesis points to a potential converging mechanism behind the pathophysiology of ASD. Additionally, the association of *SOX7* with autism has been investigated directly. A case study involving a child patient exhibiting "8p23.1 duplication syndrome", revealed a de novo 1.81 Mbp duplication event on chromosome 8 (8p23.1), spanning the region where *SOX7* lies [96]. This patient exhibited characteristic symptoms of the disorder, including delay of motor and speech development and intellectual disability, which heavily overlap with autism and related intellectual disorders. Indeed, this patient also exhibited symptoms specific to ASD, such as repetitive compulsive behavior.

A GWAS performed in a Mexican population found that *SOX7* was differentially methylated between autism cases and controls [97]. Another study also found that differential methylation was associated with an "elevated polygenic burden" for autism, and further identified that two significantly associated CpG sites were located near GWAS markers for autism on chromosome 8, in the same region as *SOX7* [53]. It is worth noting that this study also found evidence of SNPs associated with both autism and DNA methylation that were annotated to *KIZ* and *XRN2*, two genes that we also found to be significantly associated with ASD.

Changes in methylation lead to changes in gene expression, providing another plausible mechanism of *SOX7* involvement: a change in *SOX7* methylation affects the expression and thus availability of the transcription factor it encodes, which has a downstream effect on the subsequent pathways *SOX7* regulates, such as Wnt/$\beta$-catenin. Indeed, both methylation studies demonstrated a negative difference in methylation between autism cases and controls. Generally speaking, undermethylation results in a less compact 3-dimensional genome structure, allowing for greater access to the gene and an increase in expression, which we see evidence of in the higher gene expression counts in autism cases versus controls in our RNA-seq data (Fig 2a and 2b) [98–100].

Finally, altered expression of *SOX7* has been shown to play a role in the development of different types of gliomas. One study demonstrated *SOX7* to be downregulated in human glioma, allowing cancer development through upregulated Wnt/$\beta$-catenin signaling [101], whereas another study demonstrated that overexpression of *SOX7* in high-grade glioma (HGG) promoted cancer development by promoting tumor growth *via* vessel abnormalization [102]. These somewhat conflicting observations demonstrate that, due to its heavy involvement in regulating several intricately linked developmental and homeostatic functions, *SOX7* expression must be delicately balanced. Interestingly, it has also been demonstrated that there is extensive overlap of genetic risk between autism and cancer [103–106]. SOX7 expression and its interactions may provide additional support to this conjecture, particularly due to its role in vasculature development and Wnt signaling regulation.

Alterations in cerebellar function and structure have previously been implicated in autism pathophysiology [107–110]. While the cerebellum is mostly known for its role in coordination and motor function, increasing evidence demonstrates the cerebellum is also involved in a variety of social and cognitive functions as well [111–113], which adds evidence to the

idea that deficiencies in cerebellar function may be one of the many contributing factors to the highly heterogenous nature of social and cognitive symptoms of ASD. Differences in cerebellar volume have been observed between ASD and neurotypical individuals [114–117], and SOX7 has been implicated in neuronal apoptosis in the cerebellum [118]. Increased blood brain barrier permeability was identified in the cerebellum of a murine model of ASD [119], and connections between changes in blood brain barrier permeability, altered vasculature, blood flow, and oxidative stress in the brain, including the cerebellum specifically, have been reviewed as possible interrelated mechanisms for ASD [120].

The methods performed are not without limitations. Gene expression is a very dynamic process that is not only tissue dependent, but also cell type specific and varies depending developmental stage and even external factors [121–126]. Certainly, these factors affecting genetic expression means that any autism-related genes which are differentially expressed at different development stages or other varying contexts may be missed. Additionally, differential expression analysis was performed on bulk-RNA, whereas it is possible that altered gene expression between autism cases and controls is cell-type specific; knowing the specifics of the expression state of specific cell types that make up key areas of the brain have a better chance of revealing mechanisms behind autism pathogenesis as well as possibly elucidate the pathophysiology behind the vast variety of ASD subtypes. Gene-based analysis also has some limitations, the most important being the reliance on a reference population for estimating linkage disequilibrium between variants. The similarity of this reference population to the population of study is crucial to the accuracy of many gene-based analyses including those performed here. Analyses using two sample Mendelian randomization also suffer from this limitation. As a result, the extent of our findings is limited to European populations, as this was our reference of choice. Future steps include a tighter integration of DNA and RNA information as well as extensions to non-European populations that have been under-researched.

These limitations notwithstanding, the study has considerable strengths. The AT method used in the gene based GWAS study can not only integrate the good properties of sum and squared sum tests but also consider LD information among genetic variants. The heatmap of the correlation between genetic variants in SOX7 (S1 Fig) indicates that rs7005905 and rs7836366, rs10100209 and rs7836366, and rs10100209 and rs7005905 have strong positive linkage disequilibrium (LD) ($\rho > 0.5$); rs4841432 has negative LD with other variants except for rs7009920. The strong LD in SOX7 and the powerful AT method warrant our identification of the autism associated gene SOX7. The successful replications of SOX7 in the replication data, gene expression data, and the associated biological plausibility underscores the robustness of the finding of the connection between SOX7 and autism. This finding may significantly advance our understanding of the etiology of autism, open new opportunities to reinvigorate the stalling autism drug development and increase the accuracy of risk prediction of autism which makes autism early intervention and prevention being possible.

## Supporting information

**S1 Fig. Heatmap of the correlation between variants in SOX7.** SNP rs7005905 and rs7836366, rs10100209 and rs7836366, and rs10100209 and rs7005905 have strong positive linkage disequilibrium (LD) ($\rho > 0.5$); rs4841432 has negative LD with other variants except for rs7009920.
(TIF)

**S1 Table. Characteristics of participants and gene *SOX7* expression status in GSE211154 RNA-seq data.**
(DOCX)

**S2 Table. 2-SMR results between ASD (Outcome) and *SOX7* expression (Exposure).**
(DOCX)

**S3 Table. 2-SMR results between ASD (Outcome) and *SOX7* expression (Exposure) (EUR GTEx samples onl.**
(DOCX)

## Acknowledgments

The study was peer reviewed and selected as a poster presentation at the 2023 American Society of Human Genetics annual meeting. We're grateful to our peers for engaging in invaluable discussions regarding our work.

## Author contributions

**Conceptualization:** Samantha Gonzales, Na Young Choi, Prabha Acharya, Sehoon Jeong, Xuexia Wang, Moo-Yeal Lee.

**Data curation:** Samantha Gonzales, Jane Zizhen Zhao, Xuexia Wang, Moo-Yeal Lee.

**Formal analysis:** Samantha Gonzales, Jane Zizhen Zhao, Xuexia Wang.

**Funding acquisition:** Xuexia Wang, Moo-Yeal Lee.

**Investigation:** Jane Zizhen Zhao, Na Young Choi, Sehoon Jeong, Xuexia Wang, Moo-Yeal Lee.

**Methodology:** Samantha Gonzales, Jane Zizhen Zhao, Na Young Choi, Xuexia Wang, Moo-Yeal Lee.

**Project administration:** Sehoon Jeong, Xuexia Wang, Moo-Yeal Lee.

**Resources:** Sehoon Jeong, Xuexia Wang, Moo-Yeal Lee.

**Software:** Samantha Gonzales, Jane Zizhen Zhao.

**Supervision:** Sehoon Jeong, Xuexia Wang, Moo-Yeal Lee.

**Validation:** Samantha Gonzales, Jane Zizhen Zhao, Na Young Choi, Prabha Acharya, Sehoon Jeong, Xuexia Wang, Moo-Yeal Lee.

**Visualization:** Samantha Gonzales, Jane Zizhen Zhao, Na Young Choi, Prabha Acharya, Moo-Yeal Lee.

**Writing – original draft:** Samantha Gonzales, Jane Zizhen Zhao, Na Young Choi, Xuexia Wang, Moo-Yeal Lee.

**Writing – review & editing:** Samantha Gonzales, Jane Zizhen Zhao, Na Young Choi, Prabha Acharya, Sehoon Jeong, Xuexia Wang, Moo-Yeal Lee.

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
