## [Decision Letter · Decision Letter 0]

26 Feb 2024

PONE-D-23-44216SOX7: Novel Autism Associated Gene Identified by Analysis of Multi-Omics DataPLOS ONE

Dear Dr. Lee,

Thank you for submitting your manuscript to PLOS ONE. After careful consideration, we feel that it has merit but does not fully meet PLOS ONE’s publication criteria as it currently stands. Therefore, we invite you to submit a revised version of the manuscript that addresses the points raised during the review process.

We look forward to receiving your revised manuscript.

Kind regards,

Chunyu Liu

Academic Editor

PLOS ONE

“This study was financially supported by the National Institutes of Health (NCATS R44TR003491 and NIDDK UH3DK119982) and the University of North Texas (Startup).”

3. We notice that your supplementary figure is uploaded with the file type 'Figure'. Please amend the file type to 'Supporting Information'. Please ensure that each Supporting Information file has a legend listed in the manuscript after the references list.

Reviewers' comments:

Reviewer's Responses to Questions

**Comments to the Author**

1. Is the manuscript technically sound, and do the data support the conclusions?

Reviewer #1: Partly

Reviewer #2: Partly

Reviewer #3: Yes

2. Has the statistical analysis been performed appropriately and rigorously? 

Reviewer #1: Yes

Reviewer #2: Yes

Reviewer #3: Yes

3. Have the authors made all data underlying the findings in their manuscript fully available?

Reviewer #1: Yes

Reviewer #2: Yes

Reviewer #3: Yes

4. Is the manuscript presented in an intelligible fashion and written in standard English?

Reviewer #1: Yes

Reviewer #2: Yes

Reviewer #3: Yes

5. Review Comments to the Author

Reviewer #1: In the study of Gonzales et al., the authors analyzed the GWAS and transcriptome data and identified SOX7 as an ASD risk gene. However, the results could not robustly support SOX7 as a novel risk gene for ASD. My several comments as listed bellow:

1、It is cautious to state that SOX7 is a novel ASD risk gene. Is this study the first one to support the connection between SOX7 and ASD?

2、The authors did not perform any experiment to support their conclusion, how to support SOX7 as an ASD risk gene?

3、Correlation between SOX7 expression and ASD phenotypes may be needed to support the conclusion.

4、eQTL analysis is required for ASD-associated genetic variants and SOX7 expression.

Reviewer #2: The authors performed gene-based association test and DEG test flawlessly. However, there is one principal issue regarding to the GWAS set used and subsequent conclusions made: the PGC ASD 2017 and PGC ASD 2019 dataset are cumulative (i.e., the raw data used in ASD 2019 is a superset of ASD 2017, and I can confirm this with confidence as a PGC panel member). Therefore, the conclusion that SOX7 association has been found in both ASD 2017 and 2019 dataset is unsurprising and statistically expected. More importantly, these two dataset cannot be claimed as replicates for their inclusion (Page 14, results).

It is recommended that a "true" replicate of gene-based association test to be performed using an independent ASD dataset such as SPARK (Matoba et al. 2020 https://www.nature.com/articles/s41398-020-00953-9). I believe it should not take too much effort to repeat the analysis and the conclusions would be significantly consolidated.

Reviewer #3: In the study conducted by Samantha Gonzales and colleagues, utilizing gene-based analysis and two extensive GWAS datasets from 2019 and 2017, identifying five genes with a substantial link to autism spectrum disorder (ASD). Further, integrated two ASD and control postmortem brain gene expression data revealed that the SOX7, an (SRY-related HMG-box) family of transcription factors as a high-confidence risk gene upregulated in ASD patients. The design of this study is straightforward, but the description of the results needs to be significantly improved. For example, the scientific rationale of each section within the results needs to be more thoroughly explained. I have the following comments:

1. A major concern of this study is the reliance on two large GWAS datasets from the Psychiatric Genomics Consortium (PGC). The dataset used for discovery, dated 2019, is not independent of the dataset used for replication, dated 2017.

2. Please combine Figures 2a and 2b as one figure. Additionally, please describe the various panels within this figure, clarifying terms such as “all sample” and “white sample.”

3. Using gene expression counts in Figures 2a and 2b creates confusion, particularly as Figure 2b is presented on different scales, please address this for clarity.

6. PLOS authors have the option to publish the peer review history of their article (what does this mean? ). If published, this will include your full peer review and any attached files.

**Do you want your identity to be public for this peer review?** For information about this choice, including consent withdrawal, please see our Privacy Policy .

Reviewer #1: No

Reviewer #2: **Yes: ** Siwei Zhang

Reviewer #3: No

---

## [Author Response · Author response to Decision Letter 1]

23 Aug 2024

We provided Respond to Reviewers in a separate file.

---

## [Decision Letter · Decision Letter 1]

6 Oct 2024

PONE-D-23-44216R1SOX7: Autism Associated Gene Identified by Analysis of Multi-Omics DataPLOS ONE

Dear Dr. Lee,

Thank you for submitting your manuscript to PLOS ONE. I would recommend provisional acceptance of your paper. We invite you to submit a revised version of the manuscript that addresses the points raised during the review process.

We look forward to receiving your revised manuscript.

Kind regards,

Chunyu Liu

Academic Editor

PLOS ONE

Journal Requirements:

I'd like to thank the authors for their efforts in addressing the reviewers' comments. While all reviewers recommended accepting the manuscript, I found that one additional concern needs to be addressed.

The MR needs to provide evaluation of instrument strength, using F-statistics. Ideally, methods like MR-Egger should be tried as well to address pleiotropy effects. The conclusion should reflect the results of these additional analyses.

Reviewers' comments:

Reviewer's Responses to Questions

**Comments to the Author**

1. If the authors have adequately addressed your comments raised in a previous round of review and you feel that this manuscript is now acceptable for publication, you may indicate that here to bypass the “Comments to the Author” section, enter your conflict of interest statement in the “Confidential to Editor” section, and submit your "Accept" recommendation.

Reviewer #1: All comments have been addressed

Reviewer #2: All comments have been addressed

Reviewer #3: All comments have been addressed

2. Is the manuscript technically sound, and do the data support the conclusions?

Reviewer #1: Yes

Reviewer #2: Yes

Reviewer #3: Yes

3. Has the statistical analysis been performed appropriately and rigorously? 

Reviewer #1: Yes

Reviewer #2: (No Response)

Reviewer #3: Yes

4. Have the authors made all data underlying the findings in their manuscript fully available?

Reviewer #1: Yes

Reviewer #2: Yes

Reviewer #3: Yes

5. Is the manuscript presented in an intelligible fashion and written in standard English?

Reviewer #1: Yes

Reviewer #2: Yes

Reviewer #3: Yes

6. Review Comments to the Author

Reviewer #1: The authors have addressed my concerns. However, the abstract contains too much description about the background knowledge. Major results should be briefly summarized rather than listed all the results in the abstract.

The abstract is suggested to be revised before publication. I have no further comments.

Reviewer #2: There is one minor issue that needs to be amended before allowing publication: in Fig. 2, please plot all data points in addition to the box-whisker plot.

Reviewer #3: The authors have done an excellent job responding to my criticisms. I am happy to now recommend this manuscript for publication in Plos One.

7. PLOS authors have the option to publish the peer review history of their article (what does this mean? ). If published, this will include your full peer review and any attached files.

**Do you want your identity to be public for this peer review?** For information about this choice, including consent withdrawal, please see our Privacy Policy .

Reviewer #1: No

Reviewer #2: **Yes: ** Siwei Zhang

Reviewer #3: No

---

## [Author Response · Author response to Decision Letter 2]

30 Oct 2024

Reviewer #2: There is one minor issue that needs to be amended before allowing publication: in Fig. 2, please plot all data points in addition to the box-whisker plot.

Response: We thank the reviewer for the insightful suggestions. We have updated Figure 2 as suggested in the updated manuscript.

---

## [Editor Report · Decision Letter 2]

13 Feb 2025

SOX7: Autism Associated Gene Identified by Analysis of Multi-Omics Data

PONE-D-23-44216R2

Dear Dr. Lee,

We’re pleased to inform you that your manuscript has been judged scientifically suitable for publication and will be formally accepted for publication once it meets all outstanding technical requirements.

Kind regards,

Chunyu Liu

Academic Editor

PLOS ONE
---

## [Editor Report · Acceptance letter]

PONE-D-23-44216R2

PLOS ONE

Dear Dr. Lee,

I'm pleased to inform you that your manuscript has been deemed suitable for publication in PLOS ONE. Congratulations! Your manuscript is now being handed over to our production team.

Kind regards,

on behalf of

Dr. Chunyu Liu

%CORR_ED_EDITOR_ROLE%

PLOS ONE